# Ischemic stroke in PAR1 KO mice: Decreased brain plasmin and thrombin activity along with decreased infarct volume

**Efrat Shavit-Stein**[1,2]☯ *, **Ekaterina Mindel**[1,3]☯, **Shany Guly Gofrit**[2], **Joab Chapman**[1,2,3,4], **Nicola Maggio**[1,2,5]

**1** Department of Neurology, Sackler Faculty of Medicine, Tel Aviv University, Tel Aviv, Israel, **2** Department of Neurology, The Chaim Sheba Medical Center, Ramat Gan, Israel, **3** Department of Physiology and Pharmacology, Sackler Faculty of Medicine, Tel Aviv University, Tel Aviv, Israel, **4** Robert and Martha Harden Chair in Mental and Neurological Diseases, Sackler Faculty of Medicine, Tel Aviv University, Tel Aviv, Israel, **5** Sagol School of Neuroscience, Tel Aviv University, Tel Aviv, Israel

☯ These authors contributed equally to this work.
\* efrat.shavit.stein@gmail.com

## Abstract

### Background

Ischemic stroke is a common and debilitating disease with limited treatment options. Protease activated receptor 1 (PAR1) is a fundamental cell signaling mediator in the central nervous system (CNS). It can be activated by many proteases including thrombin and plasmin, with various down-stream effects, following brain ischemia.

### Methods

A permanent middle cerebral artery occlusion (PMCAo) model was used in PAR1 KO and WT C57BL/6J male mice. Mice were evaluated for neurological deficits (neurological severity score, NSS), infarct volume (Tetrazolium Chloride, TTC), and for plasmin and thrombin activity in brain slices.

### Results

Significantly low levels of plasmin and thrombin activities were found in PAR1 KO compared to WT (1.6±0.4 vs. 3.2±0.6 ng/μl, p<0.05 and 17.2±1.0 vs. 21.2±1.0 mu/ml, p<0.01, respectively) along with a decreased infarct volume (178.9±14.3, 134.4±13.3 mm³, p<0.05).

### Conclusions

PAR1 KO mice have smaller infarcts, with lower thrombin and plasmin activity levels. These findings may suggest that modulation of PAR1 is a potential target for future pharmacological treatment of ischemic stroke.

**Data Availability Statement:** All relevant data are within the manuscript and its Supporting Information files.

**Funding:** The authors received no specific funding for this work.

**Competing interests:** The authors have declared that no competing interests exist.

## Introduction

Stroke is a common disease, caused by acute deprivation of oxygen and nutrients to tissue due to ischemia or hemorrhage. The lifetime risk for stroke is 25% [1], and it is the fifth leading cause of death in the United States [2], making it a major burden on the healthcare system. Clinical presentation of stroke differs according to the affected brain region. Rapid recognition of its clinical signs is necessary for immediate treatment and consequent recover of brain tissue. Thrombolysis is the major pharmacological treatment to be administered in the early phase of the disease, up to 4.5 hours of symptoms onset [3]. Endovascular intervention is an additional therapeutical approach in selected cases [4]. Thrombolysis is achieved with the use of recombinant tissue-type plasminogen activator (rtPA) aimed to generate plasmin as part of an induced fibrinolysis process. Given in the appropriate therapeutic time window, this treatment achieves good outcome in about 33% of the patients, with 10% disability free survival rates 3–6 months following treatment [5]. The relatively limited treatment options call for further evaluation of the molecular mechanisms involved in neural damage processes following ischemia.

The coagulation proteins have been recognized as central players in the modulation of the neural function in health and disease [6, 7]. Protease activated receptor 1 (PAR1) is activated by a number of serine proteases, including thrombin plasmin and FXa [8], and is their main cell signaling mediator in the central nervous system (CNS). Following ischemia, levels of thrombin rise in correlation with infarct volume in the ischemic hemisphere [9, 10]. A rise in the levels of thrombin activity is followed by a reduction in PAR1 levels; this process dynamically increases for hours after the insult [11]. Continuous elevation of thrombin activity levels indicates that this process is initiated by coagulation and hypoxia, but is continued due to other factors [12]. The zymogen plasminogen is present in the brain under physiological conditions, and it is involved in brain development [13]. When tPA binds to fibrin in a blood clot, it converts plasminogen into plasmin. The latter is also known to induce neuronal death [14]. The chemoattractant monocyte protein 1 (MCP-1) requires processing by plasmin for its activation, and therefore, lack of functioning plasmin reduces microglia induced damage [15]. Altogether this supports the existence of a fine balance between plasmin positive effect in clot lysis and its negative cellular effects on brain tissue. Recently, we have found increased plasmin activity in the brain following ischemia [16], which correlates with an increase in infarct size. Previously, published data indicate the possible protective effects of PAR1 modulation in the setting of ischemic stroke. When PAR1 is silenced (by either PAR1 antagonists or by knockout (KO)) the volume of damaged tissue caused by transient focal ischemia is significantly reduced [17]. The protective effect of PAR1 inhibition is further supported in models of PAR1 knock down and KO [18, 19], with reduced neurological deficits, and lower stroke volume. Furthermore, in contrast to the destructive effect of high thrombin levels, low levels of thrombin are protective against ischemic brain damage [20, 21]. This protective effect disappears in PAR1 KO mice [22], suggesting that PAR1 involvement in stroke pathogenesis is part of a complex modulation process. PAR1 has been suggested as a therapeutic target for stroke and specific antagonists have been used in clinical studies [23, 24], however their effectiveness vs. side effects have not yet been well established. Furthermore, the PAR1 antagonist vorapaxar is currently contraindicated in stroke patients. Due to the absence of PAR1 on WT mice platelets, the PAR1 KO mouse model allows for the differentiation between the neural and coagulation effects of PAR1. Thus, measurements of thrombin and plasmin 24 hours following experimental ischemic stroke in this model may indicate the role of neural PAR1 on stroke severity.

In the present study, we evaluated the role of PAR1, thrombin and plasmin activity levels in neural damage following permanent focal cerebral ischemia in PAR1 KO mice. Using a novel

sensitive method for a direct quantitative measure of plasmin or thrombin activity in fresh brain slices. The infarct volume following ischemia was reduced in PAR1 KO mice together with reduced plasmin and thrombin activities. These observations may shed light on the potential role of plasmin, thrombin and PAR1 in the brain function and neurological outcome following ischemic stroke and may suggest a combined therapy targeting this pathway.

## Methods

### Animals

The study was carried out on 10-weeks-old (23–30 gr) male C57BL/6J mice (WT, Envigo, Israel) and male PAR1 KO C57BL/6J (PAR1$^{-/-}$) background mice (a generous gift of Prof. Yair Reisner, the Weizmann Institute of Science, Rehovot, Israel). Animal handling and experiments were approved and performed in accordance with the Animal Care and Use Committee of the Chaim Sheba Medical Center (#1026/10/ANIM), which adhere to the Israeli law on the use of laboratory animals and according to the ARRIVE Guidelines. All animals were maintained in a controlled animal facility at 18–22˚C and 40–60% humidity, with a photoperiod of 12 hours dark/12 hours light. Animals were allowed free access to water and food before and following surgery. Genotyping for PAR1 KO was performed by polymerase chain reaction, using WT mice as reference. During the establishment of the colony establishment, the health measures of the mice were monitored twice a week (weight, mobility, ability to move freely around the cage and the condition of the fur). Loss of a 15% of body weight was considered a humane endpoint (no weight loss was documented in this cohort). Exclusion criteria included massive bleeding and/or intracerebral hemorrhage during the procedure as described below. Two hours and 24 hours following the procedure the animals underwent a neurological evaluation by an investigator blinded to the experiment groups. Mice were scored using a five-point Neurologic Severity Scores (NSS) as described by Longa et al [25]. NSS was defined as follows: 0-no neurologic deficit; 1-failure to fully extend left forepaw; 2-circling to the left; 3-falling to the left; 4-do not walk spontaneously and have depressed level of consciousness. In line with the animal welfare guidelines in our institution, in order to minimize suffering, moribund subjects must be sacrificed. During the hours following the induction of permanent MCA occlusion, we repeatedly examined the mice and sacrificed those animals which were moribund (equivalent to a score of 4 on the NSS). These animals were therefore not available for analysis 24 hours after the procedure. Mice who had NSS of 4 were sacrificed using 100 μl pentobarbital (200 mg/ml) by intraperitoneal administration. Data of excluded animals (n = 16) were pulled for the nearest time-point NSS analysis. Number of sacrificed animals is in line with the mortality known in this model, which is approximately 20–40% [16]. Animals that survived following the procedure and were not sacrificed due to exclusion criteria (WT = 19/25; PAR1 KO = 13/23) were sacrificed following 24 hours and were randomly assigned to the experimental groups.

### Permanent middle cerebral artery occlusion (PMCAo) mouse model

PMCAo model with a monofilament suture was performed as previously described [9, 26, 27]. Anesthesia was performed with 2.5% isoflurane mixed in oxygen and delivered through a face-mask. A silicone-coated filament (Doccol Corp, Redlands, CA, USA) was inserted through a small hole in the right external carotid artery. The filament was carefully advanced up to 11 mm from the carotid artery bifurcation. Throughout the procedure, the mouse's body temperature was maintained at 37˚C with the aid of a heating pad (F.S.T., CAT# 21061–90, USA). Following surgery, 1 ml of saline was injected by subcutaneous administration into the loose skin

over the intrascapular [28]. The mice were kept in a lamp-heated chamber for 2–5 hours, according to the experimental setting [29].

## Plasmin and thrombin activity assays

Plasmin and thrombin activities in brain slices were measured as previously described [9, 16]. Following sacrifice, the brain was removed and placed in a steel brain matrix (1 mm, Coronal, Stoelting, IL, USA). First, the brain was cut into sagittal sections in its midline separating the non-ischemic and ischemic hemispheres. Then, each hemisphere was cut into 9 coronal slices (starting at slice number (no.) 3, 2 mm anterior to the bregma). Analyses were conducted on slices 3–8. The slices were placed in 96-well black microplate. *Plasmin activity*: prior to fluorescence signal reading, the plates were pre-treated by incubation while shaking for 90 minutes at 37°C in Thermo Shaker (MB100-4P), in order to allow penetration of the high molecular weight inhibitor $\alpha_2$-antiplasmin. Following this pre-treatment, a fluorometric assay measuring the cleavage of synthetic peptide substrate Boc-Val-Leu-Lys-peptidyl-7-amino-4-methylcoumaryn (AMC) (15 μM, I-1115, Bachem, Switzerland) was conducted, and defined by the linear slope of the fluorescence intensity versus time [30]. The specific plasmin activity was calculated by subtraction of the mean activity measured in the presence of $\alpha_2$-antiplasmin (3.6 mM, SRP6313, Sigma-Aldrich, Israel) from the total activity in the absence of this specific inhibitor. *Thrombin activity*: using the fluorogenic substrate, Boc-Asp(OBzl)-Pro-Arg-AMC (14 μM, I-1560, Bachem, Switzerland) a linear slope of the fluorescence intensity versus time was defined. The specific thrombin activity was measured in the presence of Bachem defined substrate buffer, 0.1% BSA, 0.1 M bestatin and 0.2 mM prolylendopeptidase inhibitor. In order to demonstrate that the measured signal represents thrombin activity, NAPAP (1 μM), a specific thrombin inhibitor, was added to the microplate. For calibration, known concentration of human plasmin and bovine thrombin (P1867, T-4648, respectively, Sigma-Aldrich, Israel) were used in duplicates in the same assay. Measurements were performed by the Infinite 2000 microplate reader (Tecan, infinite 200, Switzerland) with excitation and emission filters of 360 ±35 nm and 460±35 nm, respectively. Other parameters were kinetic cycles, 25; interval time, 2 minutes; integration time, 20 μs; lag time, 0.

## Calculation of infarct volume

Infarct volume was measured as previously described [16]. Briefly, immediately following the plasmin activity assay, brain slices were placed in 2% Tetrazolium Chloride (TTC) solution for 30 minutes. Next, the slices were fixed by 4% formaldehyde for 30 minutes and mounted on microscope slides and scanned (1,200 dpi resolution). The area of the lesion, as identified by reduced TTC staining, was manually traced and calculated using ImageJ (NIH), a Java based image processing program. The calculation included correction for brain edema by subtracting the area of normal tissue in the ischemic hemisphere from the total area of the non-ischemic hemisphere [31].

## Statistical analysis

Sample size was based on previous studies, with appropriate modifications [9]. Data are presented as dot plots and mean ±SEM. When comparing two groups, the difference was determined by two-tail unpaired t-test with Welch's correction. In the case of more than two groups we preformed one-way analysis of variance (ANOVA) with Tukey post hoc analysis. For plasmin activity and infarct analysis as a function of the distance from the ischemic core, two-way ANOVA with Sidak's post hoc analysis was performed. Statistical significance was

determined as p value <0.05. Analysis and figures preparation were performed with Graphpad Prism (Version 8.0.1, GraphPad Software Inc., San Diego, CA).

## Results

### Decreased infarct volume in PAR1 KO mice

Total infarct volume at 24 hours following ischemia was calculated by summation of infarct volumes measured in each slice, in animals with neurologic severity score (NSS)<3. In total, the NSS scores included WT = 25 and PAR1 KO = 23 mice. Animals with an NSS of 4 were excluded from the following study and are included only in Fig 1A. Though no significant differences were found in mortality rates and NSS between WT and PAR1 KO mice (NSS evaluation at 2 hours, 1.4±0.1, 1.6±0.2, respectively, p = 0.8, and at 24 hours, 1.96±0.2, 2.54±0.3,

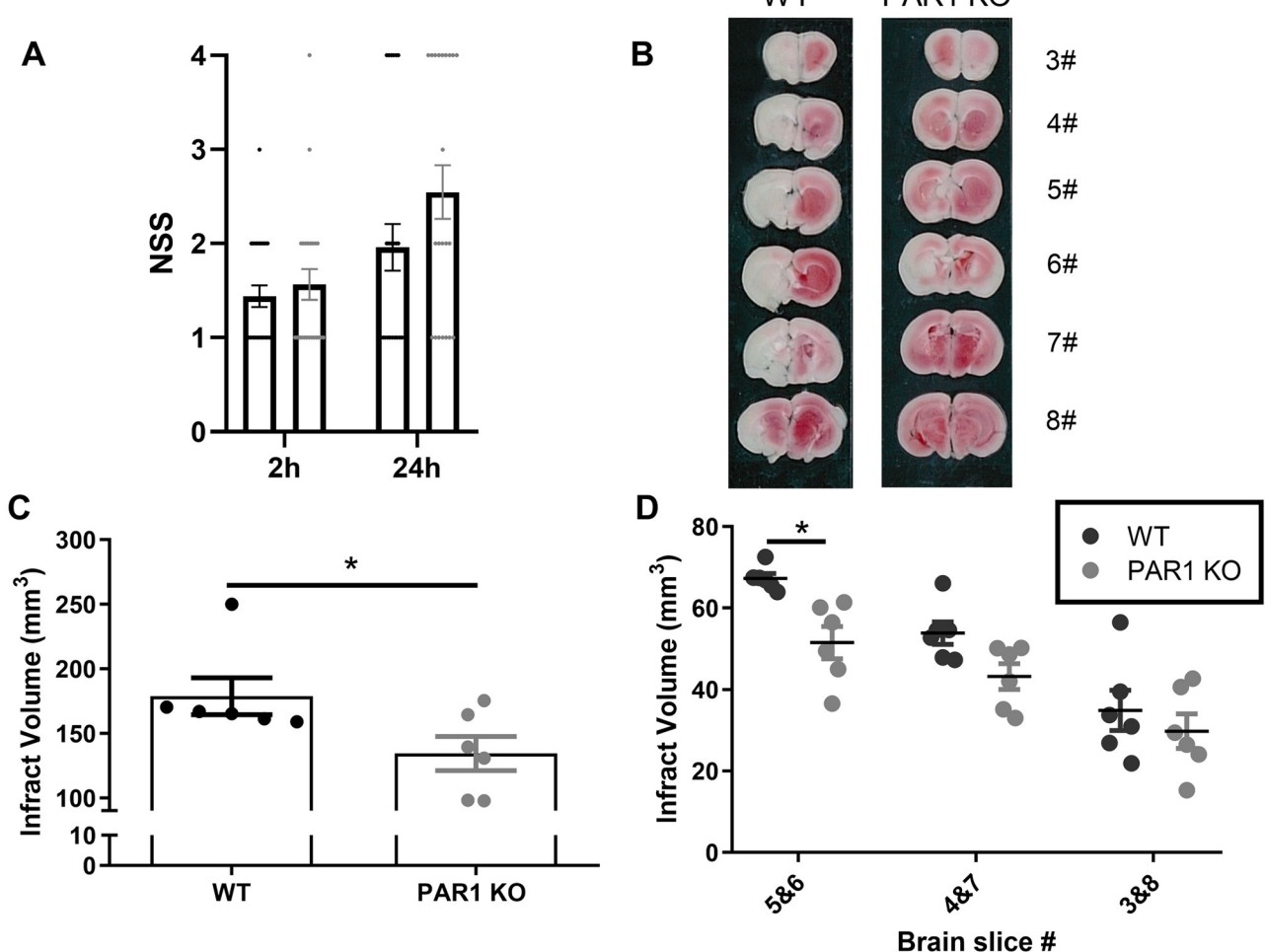

**Fig 1. Preserved NSS at 2 and at 24 hours following ischemia and decreased infarct volume in PAR1 KO mice 24 hours following ischemic stroke.** (A) There was no significant difference between NSS scores in WT (black circles) compared to PAR1 KO (gray circles) mice at 2- and 24-hours following ischemia (mixed model analysis, with Sidak's post-hoc analysis. 2 hours: WT = 25, PAR1 KO = 23, 24 hours: WT = 25, PAR1 KO = 22). (B) Representative images of TCC staining brain slices of WT (left panel) and PAR1 KO (right panel), including slices 3#-8#. (C) Total infarct volume is significantly lower in PAR1 KO mice compared to control (t-test, WT = 6, PAR1 KO = 6). (D) Infarct volume decreases as a function of distance from the ischemic core (slices #5&6). Infarct volume is lower in all PAR KO slices compared to WT mice (two-way ANOVA with Sidak's post-hoc analysis, WT = 6, PAR1 KO = 6). Results are presented as dot plot individual values and mean±SEM, *p<0.05.

p = 0.08, Fig 1A), the infarct volume was significantly lower in PAR1 KO mice compare to WT (representative TCC staining slices Fig 1B, 178.9±14.3, 134.4±13.3 mm$^3$, respectively, p<0.05, Fig 1C). Two-way ANOVA (animal type, infarct size) with repeated measures (distance from the core) was calculated. Using a post-hoc analysis for multiple comparisons showed that a difference in each separate slice compared (ischemic penumbra in slices #4&7: 43.2±3.4, 53.8 ±3.03 mm$^3$, respectively, p = 0.12; slices #3&8: 29.7±4.6, 34.9±5.4 mm$^3$, respectively, p = 0.68) and reached statistical significance in the ischemic core (slices #5&6, which corresponded to the infarct core, 51.5±3.9, 67.3±1.2 mm$^3$, respectively, p<0.05, Fig 1D).

## Decreased plasmin activity in PAR1 KO and association with infarct volume

Plasmin activity following permanent middle cerebral artery occlusion (PMCAo) was measured in each slice, and summed up separately in each hemisphere. Plasmin activity in the ischemic hemisphere was elevated in both WT and PAR1 KO mice. However, activity levels were significantly lower in the PAR1 KO mice compared to WT (1.6±0.4, 3.2±0.6 ng/µl, respectively, p<0.05, n = 6, Fig 2A). Measurements of plasmin activity levels in each slice

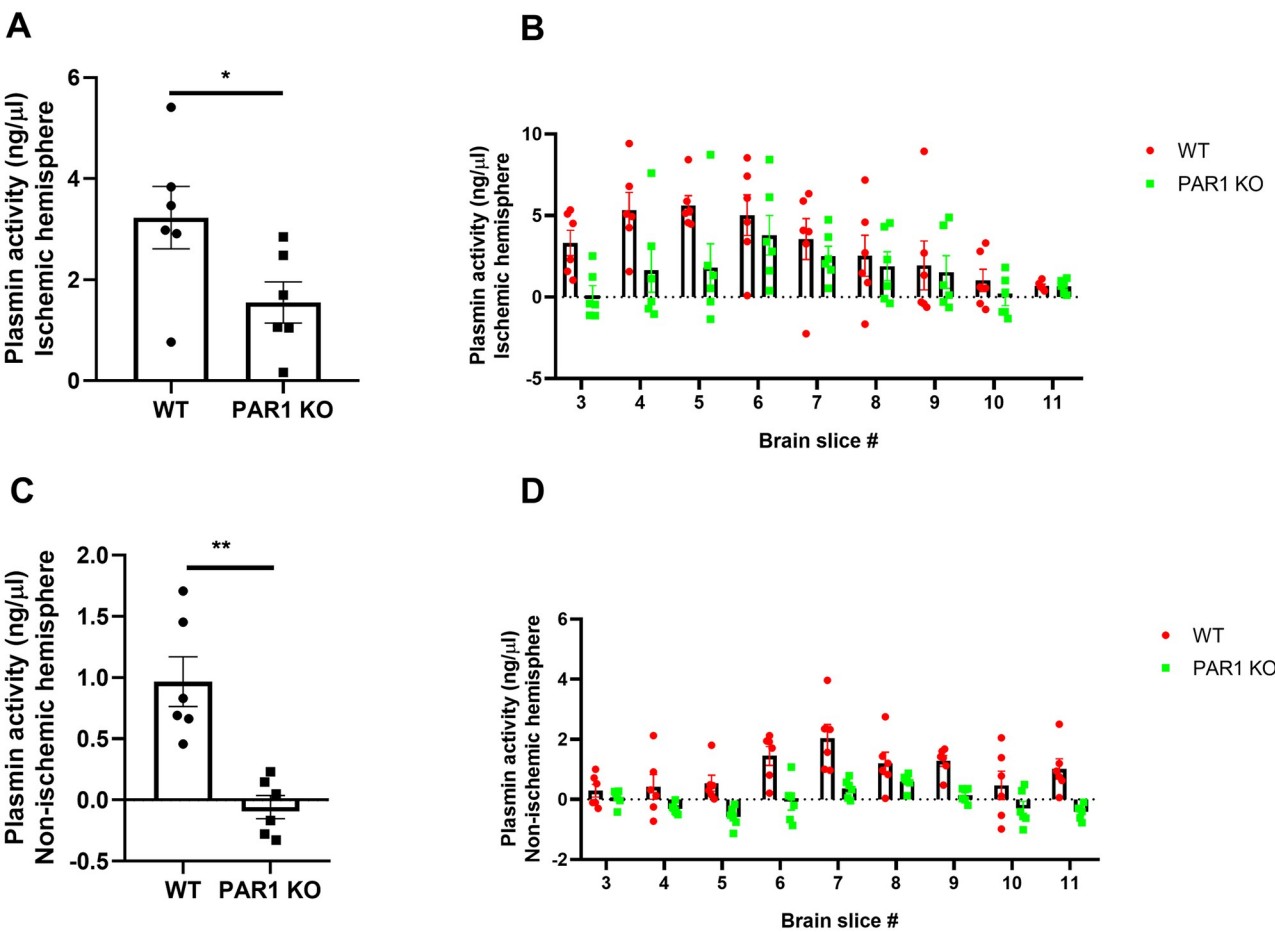

**Fig 2. Decreased plasmin activity levels in PAR1 KO mice at 24 hours following ischemia.** (A) Total plasmin activity levels were significantly lower in the ischemic hemisphere of PAR KO mice compared to WT (t-test). (B) Plasmin activity measurements by slice number show an elevation closer to the infarct core in WT and PAR1 KO mice (two-way ANOVA with Sidak's post-hoc analysis). (C) Total plasmin activity levels in the non-ischemic hemisphere were undetectable in PAR1 KO mice compared to WT (t-test). (D) Spatial dispersion was present in the non-ischemic hemisphere as well (two-way ANOVA with Sidak's post-hoc analysis). N = 6, results are presented as dot plot individual values and mean±SEM, *p<0.05, **p<0.01.

separately showed spatial dispersion, with higher levels at the ischemic core, and with lower plasmin activity levels in all the PAR1 KO slices (F(8,90) = 3.77, p<0.001 for brain slice number, and F(1,90) = 12.8 p<0.001 for mice type, Fig 2B). The non-ischemic hemisphere of PAR1 KO mice showed undetectable plasmin activity levels compared to WT mice (-0.06±0.09, 0.97 ±0.2 ng/μl, p<0.01, Fig 2C). Spatial dispersion was present in the non-ischemic hemisphere as well with no elevation opposite the ischemic core in the PAR1 KO slices (Fig 2D).

## Decreased thrombin activity in PAR1 KO mice

Thrombin activity measured in the ischemic hemisphere was significantly lower in PAR1 KO mice compared to WT (17.2±1.0, 21.2±1.0 mU/ml, p<0.01, PAR1 KO = 3, WT = 4, Fig 3A). Spatial measurements of thrombin activity were elevated in the slices at the ischemic core. Thrombin activity levels were significantly lower in PAR1 KO compare to WT mice in slice #3, and the ischemic core (slices #5 and #6) (F(8,45) = 4.688, p<0.001, Fig 3B). Thrombin

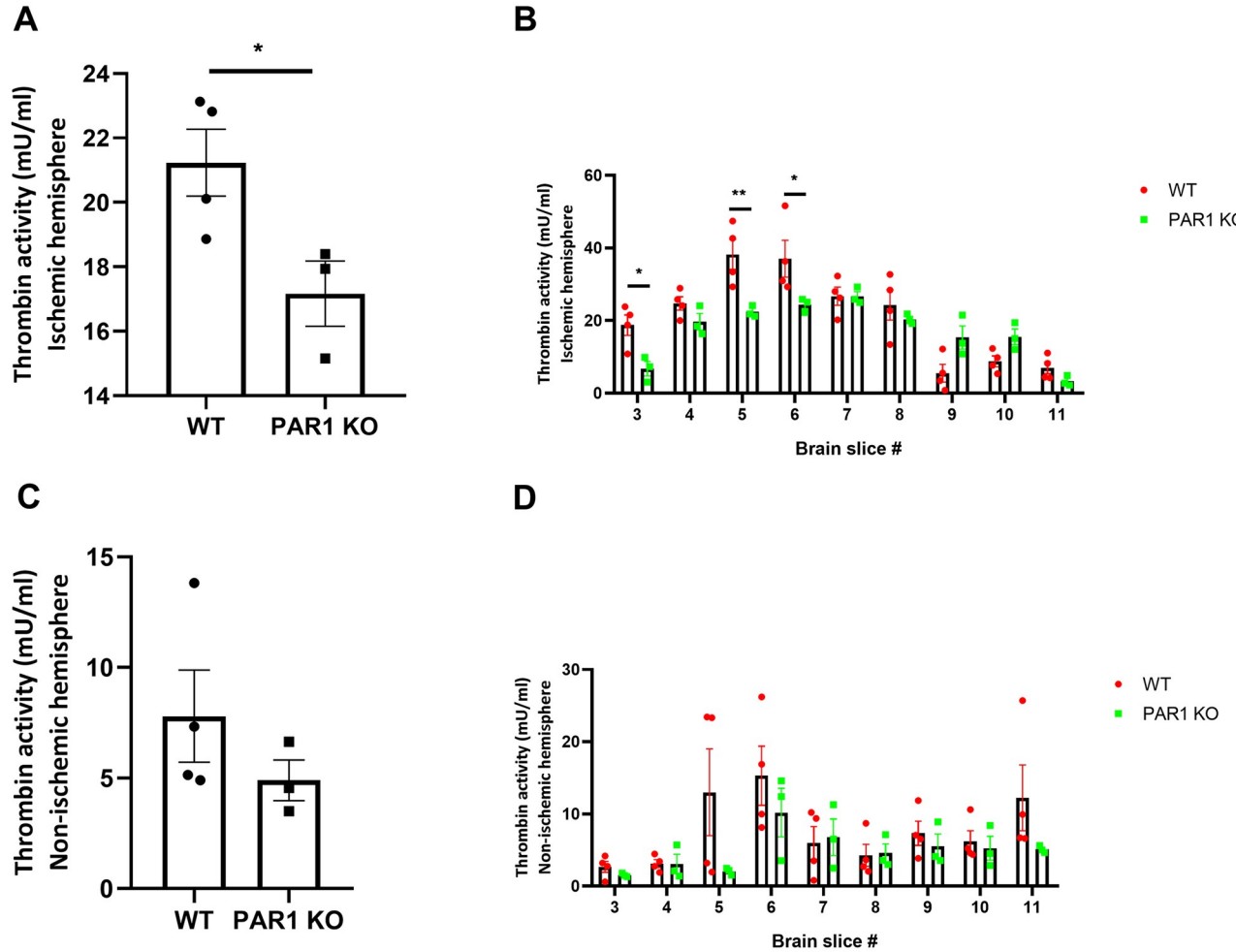

**Fig 3. Decreased thrombin activity levels in PAR KO following 24 hours following ischemia.** (A) Total thrombin activity is significantly decreased in PAR1 KO compared to WT mice (t-test). (B) Thrombin activity elevation in the ischemic hemisphere of both groups demonstrated per slice. Levels are significantly higher in WT mice in slice #3 and in slices #5 and #6 representing the ischemic core (two-way ANOVA with Sidak's post-hoc analysis). (C) A non-significant trend towards lower thrombin activity levels in PAR1 KO mice non-ischemic hemisphere (t-test). (D) Similar higher levels of thrombin activity with spatial dispersion in WT compared to PAR1 KO mice (two-way ANOVA with Sidak's post-hoc analysis). PAR1 KO = 3, WT = 4, results are presented as dot plot individual values and mean ± SEM, *p<0.05, **p<0.01.

activity showed a trend towards decrease in the non-ischemic hemisphere of the PAR1 KO as well (Fig 3C) with spatial variations (Fig 3D).

Interestingly, in the non-ischemic hemisphere the measured low levels of thrombin activity were similar between PAR1 KO and WT mice, while the measured plasmin activity was significantly lower in the PAR1 KO mice compared to WT.

## Discussion

In the present study, PAR1 KO mice show smaller infarct volumes following ischemic stroke compared to WT. This was accompanied by reduction of both thrombin and plasmin activity levels in the ischemic and non-ischemic hemispheres. PAR1 KO mice were previously reported to present with smaller infarct size, decreased brain edema and fewer behavioral impairments in response to ischemia compared to WT mice [19].

Elevation of thrombin activity levels following stroke in WT mice is well established [9, 26] and it is accompanied by a reduction of its receptor PAR1 [11] and synaptic dysfunction following reperfusion [26]. Furthermore, the known spatial differences of PAR1 expression in the brain [32], further support the findings of different thrombin activity measured in brain slices in the current study. Ischemia-induced elevation of plasmin activity levels has been recently described by our group, using a novel measuring method for plasmin activity levels in brain slices. Plasmin activity levels were found to be higher in larger infarcts, and showed a time-dependent increase [16]. The original finding of the current study is a correlation between reduced infarct size and reduced thrombin and plasmin levels in PAR1 KO mice. Elevated levels of thrombin and plasmin activity in WT mice compared to PAR1 KO during stroke are related to larger infarct size. Our results support the role of PAR1 as a key automodulator of thrombin and plasmin, which is directly related to the potential damage caused by acute ischemia.

There are number of possible explanations for the correlation between PAR KO, reduced plasmin and thrombin activity levels and smaller infarct volumes. An interesting open question is whether the absence of PAR1 acts locally in blood vessels, generating a smaller occlusion, intrinsically in the brain parenchyma, or both. Lack of PAR1 in mice platelets indicates intrinsic brain mechanism. The differences between PAR1 KO and WT can be explained by two possible mechanisms; the first, smaller infarcts in PAR1 KO leads to reduced thrombin and plasmin activity. The second, lack of PAR1 reduces thrombin and plasmin activity resulting in smaller infarcts. Our results indicate decreased levels of thrombin and plasmin in the contralateral side following stroke strengthen the second explanation which is also supported by several known cellular mechanisms.

Activation of PAR1 by high levels of thrombin leads to various neurotoxic effects via distinct mechanisms [6, 33], including microglia activation, leading to neuronal damage [34], and mediation of ischemic long term potentiation (iLTP) [12]. Selective inhibition of thrombin activity reduces CNS inflammatory markers, as seen following lipopolysaccharide injection (LPS) [35]. The latter suggests that blocking PAR1 activation may prevent some of thrombin's harmful effects and explain reduced stroke severity in these mice. The reduction in plasmin and thrombin activity levels in the PAR1 KO PMCAo model suggests that the levels of these factors are regulated by PAR1. A possible explanation may involve the astrocyte role in mediating neuroinflammation. Indeed PAR1 activates astrocytes through an IL-6 dependent pathway, resulting in a positive feedback loop, thus elevating PAR1 and thrombin levels [36]. PAR1 can be activated by both thrombin and plasmin. Plasmin leads to downstream elevation of $Ca^{+2}$ levels in astrocytes, activation of extracellular signal-regulated kinases (ERK 1/2) and potentiation of NMDA currents in pyramidal CA1 neurons [37]. A

reduced activation of mitogen-activated protein kinases was suggested as an underlying mechanism for the smaller infarct size seen in PAR1 KO mice [19]. Thrombin cleaves PAR1 at the Arg41-Ser42 site [38]. Plasmin is able to cleave and activate PAR1 at the same site at a relatively similar rate, as well as serval other sites, leading to desensitization [8]. The presence of a common binding site, together with plasmin unique activation/desensitization mode, may point towards a competitive inhibition mechanism between thrombin and plasmin, with different downstream effects.

Aside from its anti-coagulation properties, plasmin mediates neurotoxic cellular effects such as cytotoxic microglia damage [15]. Stimulation of neurons and microglia induces tPA release, which creates a vicious cycle of further microglial activation trough tPA [39]. tPA changes blood brain barrier (BBB) properties, in a dose dependent manner, which is further increased by plasminogen [40]. tPA treatment following ischemia in WT mice results in TNF alpha and IL10 elevation in the plasma, accompanied by behavioral deficits. Both inflammatory markers and behavioral deficits are reduced in mice lacking plasminogen [41]. Therefore, rtPA, which activates plasmin, may mediate positive effect in the blood vessel outside the BBB, and possible harmful effects "inside" the brain directly or indirectly on neural tissue via cellular mechanisms. Cell-surface annexin 2 (A2) and its ligand P11 accelerates plasminogen activation to plasmin by tPA. High levels of thrombin cause elevation of P11 and trafficking of ANXA2 to the cell membrane, in a PAR1 mediated mechanism [42]. Astrocytes may express uPA, which can convert plasminogen to plasmin. The presence of plasminogen may lead a PAR1-mediated proinflammatory phenotype in astrocytes [43]. This may suggest a complex feedback loop, in which PAR1 functions as a "sensor" for minor changes in thrombin levels, which results in significant fluctuation in plasmin levels. Lack of PAR1 may therefore results in a reduced plasmin production and ultimately block some of plasmin harmful effect.

This study has some limitations. Due to difficulties in breading PAR1 KO mice, colonies were relatively small, thus repetitions were limited. Although previous data indicate PAR1 KO have an improved NSS following stroke [19], in the present study this effect did not reach statistical significance. This may have several explanations: the use of a different stroke model (transient global ischemia vs. PMCAo), the temporal resolution (2 and 24 hours vs. 48 and 72 hours), and a different NSS scale (0–10 vs. 0–5). The major objective of this study was to evaluate the PAR1 plasmin thrombin interaction during the acute phase of a stroke. Therefore, a PMCAo model was used. Further evaluation at the reperfusion state should be addressed in a transient MCAo model. Measuring plasmin activity is a complex process which involves pre-treatment of the tissue including prolonged thermal shaking. This process prevents measuring thrombin activity in the same slice. Furthermore, it affects the slice volume, preventing the comparison of infarct volume in slices in which thrombin activity levels where measured. The source of plasmin activity levels in the ischemic tissue may be of brain-origin or it may originate from the plasma following BBB breakdown. Further research is needed to better define the plasmin source. Evaluation of tPA effects on PAR1 KO infarcts and plasmin levels will provide a better understanding of plasmin and PAR1 interplay. It should be stressed that a complete inhibition of PAR1 might significantly affects coagulation causing bleeding as a major side-effect (as seen with the pharmacological inhibitor vorapaxar). Therefore, intervention at this pathway should be focused on precise modulation rather than pure inhibition.

Understanding the complex interplay between PAR1 and plasmin will improve our understanding of the effects of stroke treatment, enabling the development of specific and safer pharmacological interventions.

## Supporting information

**S1 File.**
(ZIP)

## Author Contributions

**Conceptualization:** Efrat Shavit-Stein, Joab Chapman, Nicola Maggio.

**Formal analysis:** Efrat Shavit-Stein, Ekaterina Mindel, Shany Guly Gofrit, Joab Chapman, Nicola Maggio.

**Investigation:** Efrat Shavit-Stein, Ekaterina Mindel, Shany Guly Gofrit.

**Methodology:** Efrat Shavit-Stein, Joab Chapman, Nicola Maggio.

**Project administration:** Efrat Shavit-Stein, Nicola Maggio.

**Software:** Shany Guly Gofrit.

**Supervision:** Joab Chapman.

**Visualization:** Efrat Shavit-Stein, Ekaterina Mindel, Shany Guly Gofrit.

**Writing – original draft:** Efrat Shavit-Stein, Shany Guly Gofrit, Nicola Maggio.

**Writing – review & editing:** Efrat Shavit-Stein, Ekaterina Mindel, Shany Guly Gofrit, Joab Chapman, Nicola Maggio.

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
