## [Decision Letter · Decision Letter 0]

20 Jan 2021

PONE-D-20-35876

Ischemic Stroke in PAR1 KO Mice: Decreased Brain Plasmin and Thrombin Activity Together with Decreased Infarct Volume

PLOS ONE

Dear Dr. Shavit-Stein,

Thank you for submitting your manuscript to PLOS ONE. After careful consideration, we feel that it has merit but does not fully meet PLOS ONE’s publication criteria as it currently stands. Therefore, we invite you to submit a revised version of the manuscript that addresses all of the points (including new experimental data) raised by two established and one junior investigators during the review process. Please see below in the Reviewers' comments section for the detailed evaluation of your manuscript.

We look forward to receiving your revised manuscript.

Kind regards,

Vardan T. Karamyan, Pharm.D., Ph.D.

Academic Editor

PLOS ONE

Journal Requirements:

2. Please include captions for your Supporting Information files at the end of your manuscript, and update any in-text citations to match accordingly. Please see our Supporting Information guidelines for more information: http://journals.plos.org/plosone/s/supporting-information

Reviewers' comments:

Reviewer's Responses to Questions

**Comments to the Author**

1. Is the manuscript technically sound, and do the data support the conclusions?

Reviewer #1: Yes

Reviewer #2: No

Reviewer #3: Partly

2. Has the statistical analysis been performed appropriately and rigorously? 

Reviewer #1: Yes

Reviewer #2: No

Reviewer #3: No

3. Have the authors made all data underlying the findings in their manuscript fully available?

Reviewer #1: No

Reviewer #2: No

Reviewer #3: Yes

4. Is the manuscript presented in an intelligible fashion and written in standard English?

Reviewer #1: Yes

Reviewer #2: No

Reviewer #3: Yes

5. Review Comments to the Author

Reviewer #1: 1. The study has been presented in a very organized manner and it was very easy to follow however, it would be better if there is consistency for naming ischemic hemisphere (only ipsilateral or only ischemic hemisphere through out the article).

2. The total number of mice used for the study as well as individual n number for different analysis need to be included.

3. Excluding n=16 mice from study seems quite high so number of animals used for NSS needs to be specified.

4. Why the NSS study was not performed at 24h time point rather that 2 hours post stroke time point as all the activity assay was done at 24 hours post stroke.

5. For both Plasmin and Thrombin activity assay results, in case of WT the left and right hemisphere results are substantially different in fact could be statistically significant. It needs substantial explanation for such a difference.

6. The author showed spatial difference in plasmin and thrombin activity, So is there any study showing the spatial difference in the expression of PAR1 expression in brain?

7. The author did a great job explaining the importance and limitation of the study in the introduction and discussion part.

Reviewer #2: This manuscript investigates the role of PAR-1 in a mouse model of ischemic stroke. The main conclusion is that the activities of plasmin and thrombin are lower in the infarct regions of PAR-1-KO mice compared to WT mice and that this correlates with smaller infarct size in the PAR-1-KO mice.

1. In the abstract and introduction (pages 2 and 3) it is stated that PAR-1 is activated by thrombin and plasmin. It would be correct to say that PAR-1 is activated/ inactivated by many proteases that are likely to be involved in ischemic stroke.

2. Correct nomenclature and reference to the PAR-1-KO mice should be given on page 5.

3. In the methods section there is a considerable overlap in the description of plasmin and thrombin activity assays. It appears that they are essentially the same except for the substrate. It may be easier to give a composite description for both and point out the differences (page 6 and 7).

4. Substrate concentration and details are missing in the description of plasmin assay (page 6).

5. From the discussion about these assays on page 14 it is clear that the methodological description of these assays is completely inadequate. "Pretreatments" and sequential measurement of plasmin and thrombin activity should be described correctly and in more detail.

6. The logic behind the use of permanent ischemia versus transient ischemia should be described in detail (page 14).

7. TTC staining images should be shown.

8. The discussion is quite lengthy and touches on many points points outside the scope of the current study. It should focus more closely on how these results (might) explain the differences observed in PAR-1-KO mice.

9. Fig. 1: (i) NSS score at 24 h should also be shown. (ii) Number of animals in each group should be indicated in the figure legend. Which statistical test was used in each of the panels should be indicated in the figure legend. (iii) The statement related to description of panel C does not seem to be statistically tested.

10. please apply the above criteria to the other manuscript figures/ legends.

11. It is not clear why plasmin activity is negative in some samples (Figure 2).

12. Supposedly, specific substrates for plasmin and thrombin have been used to measure enzyme activity. Their specificity is supported by the use of anti-plasmin and NAPAP respectively. However, neither the substrate, nor the inhibitor, are specific for the said enzymes. Inhibitory antibodies would be a more suitable choice to demonstrate specificity of these assays.

Reviewer #3: Ischemic stroke is a major health issue that has limited treatment options. PAR1 is activated by thrombin and plasmin and has important signaling roles in the central nervous system. The report by Shavit-Stein is a correlational study that examines infarct size, plasmin activity, and thrombin activity in an ischemic stroke model in wild type and PAR1 knock out mice. The PAR1-KO animals had a decrease in infarct size. Brain slices from the PAR1-KO mice had a decrease in thrombin generation and plasmin generation compared to wild type mice.

The reported decreases in infarct size, thrombin and plasmin are ~20%, 20%, and 50%, respectively. The authors should comment on the biological significance of these differences.

Are the differences in these three parameters expected to be a result of the injury? or a physiological response to the injury? The authors have focused on the neuronal cells. However, other cells may also contribute to the observations. Identifying which cells are driving the phenotype will aid in developing hypothesis for determining the underlying mechanisms.

The manuscript would be significantly strengthened with additional experiments that link the three observations (infarct size, thrombin, and plasmin) or provide potential mechanisms for how they could be related in this stroke model.

The authors should clearly state the number of animals used in each experiment. It appears to be 3-6. What is the required number of animals for these studies to determine true differences (power analysis).

The authors highlight that the mouse model allows them to examine PAR1's impact on the CNS without a contribution of platelets due to differences in PAR expression across species. This is a valuable tool, however the impact of interfering with PAR1 signaling in humans should be addressed in the Discussion. Given that the PAR1 antagonist (vorapaxar) is contraindicated for stroke, this is particularly relevant.

6. PLOS authors have the option to publish the peer review history of their article (what does this mean?). If published, this will include your full peer review and any attached files.

Reviewer #1: No

Reviewer #2: No

Reviewer #3: No

---

## [Author Response · Author response to Decision Letter 0]

2 Feb 2021

Reviewer #1:

1. The study has been presented in a very organized manner and it was very easy to follow however, it would be better if there is consistency for naming ischemic hemisphere (only ipsilateral or only ischemic hemisphere through out the article).

We thank the reviewer for this comment. We have edited the text and results accordingly

2. The total number of mice used for the study as well as individual n number for different analysis need to be included.

This was included in the relevant results sections and figure legends. 

3. Excluding n=16 mice from study seems quite high so number of animals used for NSS needs to be specified.

The number of animals used for NSS was added to the manuscript (page 6, lines 110-114)

4. Why the NSS study was not performed at 24h time point rather that 2 hours post stroke time point as all the activity assay was done at 24 hours post stroke.

The NSS at 24h was added to the text and the Figure (Page 8 line 175, Figure 1, and Figure 1 legend page 9 lines 185-189)

5. For both Plasmin and Thrombin activity assay results, in case of WT the left and right hemisphere results are substantially different in fact could be statistically significant. It needs substantial explanation for such a difference.

The reviewer is right and this is a main statement already described by our group in several published papers. The brain intrinsic thrombin concentration significantly increases following MCAo as it was described by Bushi et al (Bushi et al. 2017 Frontiers in neurology) so it does the brain intrinsic plasmin concentration as described by Mindel et al (Mindel et al. 2020, journal of neuroscience research). This is an important point and it was further highlighted into the manuscript (page 12 line 248, 253-254). Additional new references were added to the text. 

6. The author showed spatial difference in plasmin and thrombin activity, So is there any study showing the spatial difference in the expression of PAR1 expression in brain?

We thank the reviewer for this important comment. PAR1 spatial expression was described in the literature (SF Traynelis Experimntal Neurology 2004) and this description was added to the manuscript (page 12 lines 250-251). 

7. The author did a great job explaining the importance and limitation of the study in the introduction and discussion part.

Reviewer #2: 

This manuscript investigates the role of PAR-1 in a mouse model of ischemic stroke. The main conclusion is that the activities of plasmin and thrombin are lower in the infarct regions of PAR-1-KO mice compared to WT mice and that this correlates with smaller infarct size in the PAR-1-KO mice.

1. In the abstract and introduction (pages 2 and 3) it is stated that PAR-1 is activated by thrombin and plasmin. It would be correct to say that PAR-1 is activated/ inactivated by many proteases that are likely to be involved in ischemic stroke.

The reviewer is right and the manuscript was edited accordingly (page 2, line 21, page 3 lins 49-50). 

2. Correct nomenclature and reference to the PAR-1-KO mice should be given on page 5.

The nomenclature was clarified in the methods section (page 5 line 88). 

3. In the methods section there is a considerable overlap in the description of plasmin and thrombin activity assays. It appears that they are essentially the same except for the substrate. It may be easier to give a composite description for both and point out the differences (page 6 and 7).

Indeed, the two methods are similar. The Method section was changed accordingly (page 6-7, lines 125-149). Both methods have been published previously (Bushi 2013, Mindel 2020). 

4. Substrate concentration and details are missing in the description of plasmin assay (page 6).

The information was added to the methods section (page 7 lines 135, 140). 

5. From the discussion about these assays on page 14 it is clear that the methodological description of these assays is completely inadequate. "Pretreatments" and sequential measurement of plasmin and thrombin activity should be described correctly and in more detail.

We thank the reviewer for pointing this out. More details regarding the pretreatment of the tissue were added to the methods (pages 6-7, lines 131-134, page 14, lines 311-312). 

6. The logic behind the use of permanent ischemia versus transient ischemia should be described in detail (page 14).

We agree with the reviewer that both models represent clinical manifestations of an ischemic stroke. However in this study we decided to focus on the changes occurring during the acute ischemic phase. This phase is much better represented by the permanent ischemia model. The transient ischemia model focuses on reversible ischemic changes due to reperfusion. Therefore, we decided to establish our findings on a permanent model prior to evaluate changes in a transient ischemia model. This was better explained in the discussion (page 14-15, lines 308-310). 

7. TTC staining images should be shown.

Representative TTC staining images were added to Figure 1 (Figure 1B). 

8. The discussion is quite lengthy and touches on many points points outside the scope of the current study. It should focus more closely on how these results (might) explain the differences observed in PAR-1-KO mice.

The discussion was polished accordingly and it is now more focused to the discussion of the obtained results.

9. Fig. 1: (i) NSS score at 24 h should also be shown. (ii) Number of animals in each group should be indicated in the figure legend. Which statistical test was used in each of the panels should be indicated in the figure legend. (iii) The statement related to description of panel C does not seem to be statistically tested.

Fig 1 (i) figure 1 was edited and now includes NSS at 24 h as well.

(ii) number of animals was added to the legend, as well as statistical analysis used. 

An explanation regarding statistical evaluation in panel C was added to the Results. 

10. please apply the above criteria to the other manuscript figures/ legends.

Animal numbers and statistical analyses were added to all legends. 

11. It is not clear why plasmin activity is negative in some samples (Figure 2).

Plasmin activity measurement is based on subtraction of averaged activity measured in the presence of plasmin inhibitor from the total activity measured in the corresponding brain slice from each individual tested mouse. In some individuals, the total activity measured was slightly lower than the average activity in the presence of plasmin inhibitor. That is the reason for which some individual mice brain plasmin calculated activity is negative. The interpretation of such negative values is undetectable for plasmin specific activity. This was in-depth studied in previous publication from our group (Mindel et al. 2020 Journal of Neuroscience Research). 

12. Supposedly, specific substrates for plasmin and thrombin have been used to measure enzyme activity. Their specificity is supported by the use of anti-plasmin and NAPAP respectively. However, neither the substrate, nor the inhibitor, are specific for the said enzymes. Inhibitory antibodies would be a more suitable choice to demonstrate specificity of these assays.

Both substrates are known to be relatively specific to the studied enzyme (as can be seen in their catalog description and previous publishes). NAPAP and anti-plasmin are known as thrombin and plasmin specific inhibitors, respectively. In our hands, those two inhibitors resulted in a variety of scientific works (Mindel 2020, Bushi 2015, 2016, 2017, Gera 2018, 2019, Shavit-Stein 2011). 

Reviewer #3:

 Ischemic stroke is a major health issue that has limited treatment options. PAR1 is activated by thrombin and plasmin and has important signaling roles in the central nervous system. The report by Shavit-Stein is a correlational study that examines infarct size, plasmin activity, and thrombin activity in an ischemic stroke model in wild type and PAR1 knock out mice. The PAR1-KO animals had a decrease in infarct size. Brain slices from the PAR1-KO mice had a decrease in thrombin generation and plasmin generation compared to wild type mice.

The reported decreases in infarct size, thrombin and plasmin are ~20%, 20%, and 50%, respectively. The authors should comment on the biological significance of these differences.

The results support for the role of PAR1 as a key automodulator of thrombin and plasmin production which have deleterious effects on the brain tissue leading to increased infarct volume. In WT mice during ischemic stroke thrombin levels in the brain increase more than tenfold (Bushi D. 2017, Frontiers in Neurology) where under similar conditions brain plasmin levels are increased by 2-3 fold (Mindel et al. 2020 Journal of Neuroscience Research). The more pronounced effect on brain plasmin levels may be a result of a complex interaction with PAR1 and thrombin possibly aimed at counteracting harmful effects. (page 12, line 257-259)

Are the differences in these three parameters expected to be a result of the injury? or a physiological response to the injury? The authors have focused on the neuronal cells. However, other cells may also contribute to the observations. Identifying which cells are driving the phenotype will aid in developing hypothesis for determining the underlying mechanisms.

The specific cellular expression of this pathway components is indeed highly important in order to shed light on the possible underlying mechanism. Indeed, not only neurons play a role in the thrombin/plasmin/PAR1 pathway. Glia cells may be an even more important player. 

A major pathway involved in this interaction is brain inflammation mediated by astrocytes. The measurements that we performed at 24hrs following the stroke are in line with the time frame in which inflammation peaks. 

Astrocytes are known to produce uPA that convers plasminogen to plasmin which further leads to proinflammatory cytokines production in a PAR1-dependent mechanism (J Neuroinflammation. 2019 Dec 6;16(1):257. doi: 10.1186/s12974-019-1657-3. Fibrinolysis protease receptors promote activation of astrocytes to express pro-inflammatory cytokines). 

This interaction has been previously studied in spinal cord injury model which shows proinflammatory cytokines secreted by astrocytes has autocrine effect enhances thrombin secretion in PAR1 dependent manner (Targeting the Thrombin Receptor Modulates Inflammation and Astrogliosis to Improve Recovery after Spinal Cord Injury, neurobiology of disease 2016)

Our results showing reduced thrombin secretion in PAR1 KO mice in the stroke model are compatible with the PAR1 mediated proinflammation processes. 

This was added to the manuscript (pages 13, 14, lines 275-278, 198-299). 

The manuscript would be significantly strengthened with additional experiments that link the three observations (infarct size, thrombin, and plasmin) or provide potential mechanisms for how they could be related in this stroke model.

As stated in the previous points, the relatively complex stroke model in whole brain suggests a role of PAR1 in determining thrombin and plasmin levels. A previous study has found that thrombin acts through PAR1 to regulate levels of annexin A2 which is a key determinant of plasmin generation (Peterson 2003 and Pontecorvi 2019). This is now better explained in the revised discussion (page 14, lines 295-299).

The authors should clearly state the number of animals used in each experiment. It appears to be 3-6. What is the required number of animals for these studies to determine true differences (power analysis).

exact number of animals used was added to the methods (page 5-6 lines 105-114) as well as the relevant paragraphs in the results section and figure legends. 

The authors highlight that the mouse model allows them to examine PAR1's impact on the CNS without a contribution of platelets due to differences in PAR expression across species. This is a valuable tool, however the impact of interfering with PAR1 signaling in humans should be addressed in the Discussion. Given that the PAR1 antagonist (vorapaxar) is contraindicated for stroke, this is particularly relevant.

We thank the reviewer for this insightful comment. Our results suggest that blocking PAR1 may have a beneficial effect in stroke. However, blocking PAR1 in humans by vorapaxar is currently contraindicated in stroke due to bleeding risks. A selective blocker of PAR1 activation in the brain would be a promising therapeutic target. Indeed, PAR1 activation by aPC may have protective neurological outcomes (Maggio, 2014 hippocampus). Our group is currently studying various options for the modulation of PAR1 to bring novel selective compounds for the potential treatment of stroke in humans (page 15, lines 317-320).

---

## [Decision Letter · Decision Letter 1]

26 Feb 2021

Ischemic Stroke in PAR1 KO Mice: Decreased Brain Plasmin and Thrombin Activity Along with Decreased Infarct Volume

PONE-D-20-35876R1

Dear Author,

We’re pleased to inform you that your manuscript has been judged scientifically suitable for publication and will be formally accepted for publication once it meets all outstanding technical requirements.

Kind regards,

Vardan Karamyan, Pharm.D., Ph.D.

Academic Editor

PLOS ONE

Additional Editor Comments (optional):

Reviewers' comments:

Reviewer's Responses to Questions

**Comments to the Author**

1. If the authors have adequately addressed your comments raised in a previous round of review and you feel that this manuscript is now acceptable for publication, you may indicate that here to bypass the “Comments to the Author” section, enter your conflict of interest statement in the “Confidential to Editor” section, and submit your "Accept" recommendation.

Reviewer #1: All comments have been addressed

Reviewer #2: All comments have been addressed

2. Is the manuscript technically sound, and do the data support the conclusions?

Reviewer #1: Yes

Reviewer #2: Yes

3. Has the statistical analysis been performed appropriately and rigorously? 

Reviewer #1: Yes

Reviewer #2: Yes

4. Have the authors made all data underlying the findings in their manuscript fully available?

Reviewer #1: Yes

Reviewer #2: Yes

5. Is the manuscript presented in an intelligible fashion and written in standard English?

Reviewer #1: Yes

Reviewer #2: Yes

6. Review Comments to the Author

Reviewer #1: (No Response)

Reviewer #2: (No Response)

7. PLOS authors have the option to publish the peer review history of their article (what does this mean?). If published, this will include your full peer review and any attached files.

Reviewer #1: **Yes: **Abdullah Al Shoyaib

Reviewer #2: No

---

## [Editor Report · Acceptance letter]

4 Mar 2021

PONE-D-20-35876R1 

Ischemic Stroke in PAR1 KO Mice: Decreased Brain Plasmin and Thrombin Activity Along with Decreased Infarct Volume 

Dear Dr. Shavit-Stein:

I'm pleased to inform you that your manuscript has been deemed suitable for publication in PLOS ONE. Congratulations! Your manuscript is now with our production department. 

Kind regards, 

on behalf of

Dr. Vardan Karamyan 

Academic Editor

PLOS ONE